# Toll-Like Receptors (TLRs), NOD-Like Receptors (NLRs), and RIG-I-Like Receptors (RLRs) in Innate Immunity. TLRs, NLRs, and RLRs Ligands as Immunotherapeutic Agents for Hematopoietic Diseases

**DOI:** 10.3390/ijms222413397

**Published:** 2021-12-13

**Authors:** Katarzyna Wicherska-Pawłowska, Tomasz Wróbel, Justyna Rybka

**Affiliations:** Department and Clinic of Hematology, Blood Neoplasms and Bone Marrow Transplantation, Wroclaw Medical University, 50-367 Wroclaw, Poland; tomasz.wrobel@umw.edu.pl (T.W.); justyna.rybka@umw.edu.pl (J.R.)

**Keywords:** innate immunity, Toll-like receptors, NOD-like receptors, RIG-I-like receptors, immunotherapy, hematopoietic diseases

## Abstract

The innate immune system plays a pivotal role in the first line of host defense against infections and is equipped with patterns recognition receptors (PRRs) that recognize pathogen-associated molecular patterns (PAMPs) and damage-associated molecular patterns (DAMPs). Several classes of PRRS, including Toll-like receptors (TLRs), NOD-like receptors (NLRs), and RIG-I-like receptors (RLRs) recognize distinct microbial components and directly activate immune cells. TLRs are transmembrane receptors, while NLRs and RLRs are intracellular molecules. Exposure of immune cells to the ligands of these receptors activates intracellular signaling cascades that rapidly induce the expression of a variety of overlapping and unique genes involved in the inflammatory and immune responses. The innate immune system also influences pathways involved in cancer immunosurveillance. Natural and synthetic agonists of TLRs, NLRs, or RLRs can trigger cell death in malignant cells, recruit immune cells, such as DCs, CD8+ T cells, and NK cells, into the tumor microenvironment, and are being explored as promising adjuvants in cancer immunotherapies. In this review, we provide a concise overview of TLRs, NLRs, and RLRs: their structure, functions, signaling pathways, and regulation. We also describe various ligands for these receptors and their possible application in treatment of hematopoietic diseases.

## 1. Introduction

A properly-functioning human immune system is an essential element in maintaining systemic homeostasis. It is responsible for recognizing and controlling infections caused by invasion of pathogenic microorganisms. Immunity is divided into innate and acquired, while non-specific immunity mechanisms constitute the first line of defense of the human organism against pathogens entering the system. Acquired immunity is responsible for fighting infection in its later stages and for producing immune memory [1,2]. The mechanisms of innate immunity activate a specific immune response directed precisely toward the microorganism that caused the disease and toward infected host cells [2].

Pattern recognition receptors (PRRs) play a major role in innate immunity. This is a large group of receptors characterized by several common features, such as recognition of pathogen-associated molecular patterns (PAMPs), continuous expression in the host independent of past infections, and detection of microorganisms regardless of their developmental and life cycle stage. These receptors are localized on the surface or inside of macrophages/monocytes, dendritic cells, NK cells, mast cells, neutrophils, and eosinophils, but also on non-specialized, non-immune epithelial cells, endothelial cells, or fibroblasts [3,4]. PAMPs are building blocks of microorganisms that are essential for their survival and thus are difficult to alter in the process of escaping the host’s immune mechanisms. They involve the wall structure of bacteria, fungi, genetic material, and many other substances recognized by specific receptors [5]. In addition to PAMPs, PRRs also recognize damage-associated molecular patterns (DAMPs) that are released from damaged host cells and tissues. DAMPs include hyaluronic acid, histones, mRNAs, cholesterol crystals, and others [5,6,7].

Six groups of PRRs have been described and they include: Toll-like receptors (TLRs), retinoic acid-inducible gene I (RIG-I)-like receptors (RLRs), nucleotide-binding domain and leucin-rich repeat-containing receptors (NLRs), C-type lectin receptors (CLRs), cyclic AMP-GMP) synthase and stimulator of interferon genes (cGAS-STING pathway), and AIM2-like receptors (ALRs) [8,9,10,11,12]. TLRs and CLRs are transmembrane receptors that are found both on the cell surface and in intracellular membranes, i.e., the endoplasmic reticulum or lysosome wall. The intracellular receptor group includes RLRs, NLRs, and cGAS-STING pathway. Stimulated receptors trigger a cascade of signaling pathways leading to increased production of interferon I (IFN-I), proinflammatory cytokines (interleukin-1—IL-1; interleukin-6—IL-6; tumor necrosis factor—TNF), and other proinflammatory substances. The substances released in this process cause an increased influx of immune cells and activation of the nonspecific response system. In addition to stimulating the production of IFN-I and proinflammatory cytokines, another way to fight infection is through direct destruction of infected cells by pyrolysis or autophagocytosis [1,3]. Once DAMPs are recognized, signaling pathways within the cells are also activated, leading to the production of various cytokines and chemokines that ultimately result in the induction of so-called “sterile inflammation.” This process plays an important role in the repair and regeneration of damaged tissues [6,7].

Hematopoietic diseases are characterized by heterogeneity and include leukemias, myelodysplastic syndromes, myeloproliferative syndromes, Hodgkin’s lymphoma, and a large group of non-Hodgkin’s lymphomas (NHL). Currently, numerous studies are underway to develop new therapies that will achieve high response rates, long-term remissions, and improve overall survival of patients with hematologic malignancies, especially in patients who are initially chemo resistant. The direction of this search is mainly focused on immunotherapy and the combination of immunotherapy with conventional treatments [13,14,15].

In this article, we present a description of the structure, functions, signaling pathways, ways of regulation, and potential use in the treatment of hematopoietic proliferative diseases of three most important groups of PRRs: TLRs, NLRs, and RLRs.

## 2. Toll-Like Receptors (TLRs)

### 2.1. Structure, Location, and Functions

Toll-like receptors are the first group included in the broad family of PRRs. The first receptors encoded by the mutated *toll* gene were described in fruit flies (*Drosophila melanogaster*), in which they are responsible for developmental processes and the immune system [16,17]. The discovery of receptors responsible for detecting and controlling fungal infections in fruit flies set off a wave of research looking for similar receptors in other species. Molecular and cytogenetic studies have identified TLRs among all living multicellular organisms that face microbial attack. Toll-like receptors are germline encoded, evolutionarily old proteins [18,19,20]. Each species has a specific, genetically determined number of TLRs—from 9 in *Drosophila* to 222 in purple sea urchin [21]. In mammals, 13 genes encoding TLRs have been identified so far (TLR1-TLR10 in humans and TLR1-TLR13, but without TLR10 in mice) [3,22].

Toll-like receptors are expressed on many cells, both in the immune system—macrophages, dendritic cells (DCs), B cells, NK cells, some T cells, as well as on the surface of epithelial and endothelial cells and fibroblasts. Some TLRs are localized to the cell surface (TLRs 1, 2, 4, 5, 6) and the remaining group of TLRs (TLRs 3, 7, 8, 9) are bound to endosomal membranes intracellularly [3,4,22].

All TLRs show a similar domain organization as they are members of the trans-membrane protein I family. TLRs are composed of an extracellular N-terminal leucine-rich repeat (LRR), a single transmembrane domain, and an intracellular Toll/IL-1R receptor, known as a cytoplasmic TIR domain. The LRR domain consists of 16–28 tandem repeats of the LRR motif and is responsible for recognizing ligands such as proteins (e.g., bacterial flagellin), sugars (e.g., fungal zymogen), lipids (bacterial lipopolysaccharide), and nucleic acids (DNA and RNA of viruses). The intracellular domain of TIR consists of approximately 150 amino acids and shows similarity to the cytoplasmic region of the IL-1 receptor. It is essential for activating the signal transduction cascade [12,23,24,25].

In mammals, Toll-like receptors are synthesized in the endoplasmic reticulum (ER) and then transported to the target site. The endoplasmic reticulum-associated chaperone proteins gp96, PRAT4A, and Unc93B1 play important roles in the generation, maturation, and proper folding of receptors [26,27,28]. In cells lacking gp96 and PRAT4A, TLRs receptors (except TLR3) do not show their ligand recognition activity leading to a significant reduction in the production of proinflammatory cytokines, IFN I and other chemokines in response to infection [29,30]. Unc93B1 protein is a chaperone protein that binds endoplasmic TLRs (TLR 3, 7, 9) and TLR5 in the cytoplasmic reticulum, facilitating their maturation and proper folding. Once bound to a receptor, it remains bound to it and is essential for maintaining a stable receptor conformation. Unc93B1 accelerates receptor attachment to endosome membranes dissociating from the ER [31,32].

Upon attachment to the endosomal membrane, the LRR domain of TLRs 3, 7, 8, 9, 13 receptors is fragmented by cathepsins. The resulting domains remain connected to each other, allowing receptor dimers to form and function properly. Interestingly, intact receptors can still recognize PAMPs but are unable to trigger the signaling cascade and pass on the threat recognition information [33,34,35,36,37].

Different TLRs recognize different PAMPs and DAMPs in the body. TLR1, TLR2, and TLR6 receptors form heterodimers (TLR1/2, TLR2/6), through which they recognize mainly triacylated lipopeptides from Gram-negative bacteria and diacylated lipopeptides from *Mycoplasma* spp. [38,39] TLR2 itself presents the widest range of detectable ligands, and can respond to the presence of, among others, bacterial proteins (e.g., V-antigen from *Yersinia*), hemagglutinins from smallpox virus, glycolipids, glycopeptides, and lipoproteins from *E. coli*, *B. burgdorferi*, *M. tuberculosis* [40,41]. When combined with a protein known as dectin-1 (a member of the C-type lectin family), it recognizes zymosan [42]. TLR3 recognizes viral double-stranded RNA (dsRNA) from, for example, reoviruses. In addition, it recognizes dsRNAs arising during replication of single-stranded RNAs (ssRNAs) of viruses, e.g., *West Nile virus, RSV*, or *EMCV* (encephalomyocarditis virus). The ligand for TLR3 can also be a synthetic dsRNA analog called poly(I:C) [43,44,45]. TLR4 recognizes lipopolysaccharide (LPS) derived from the wall of Gram-negative bacteria. For proper recognition of LPS, TLR4 requires the formation of a dimer with the membrane protein MD2 [46]. In addition, TLR4 recognizes glycosaminophospholipids from *Trypanosoma*, fusion proteins from *RSV* and also envelope proteins from mouse mammary tumor virus (*MMTV*) [47,48,49,50]. In addition, it indirectly or directly detects DAMPs such as heat shock proteins, fibrinogen, hyaluronic acid, beta-defensins, and others [51]. TLR5 recognizes monomers of flagellin, which is a structural protein of bacteria that have the ability to move. TLR5 is abundant on respiratory epithelial cells and in the lamina propria of the small intestine, as detection of bacterial ligands is important in counteracting the adhesion and invasion of microorganisms that cause respiratory tract infections and gastrointestinal infections [52,53]. TLR7, TLR8, and TLR9 are localized to the cell wall of endosomes where they detect nucleic acids, ssRNA (TLR7 and TLR8) [54,55], and unmethylated CpG containing ssDNA (TLR9) from viruses and bacteria [56,57]. Ligands for the human TLR10 receptor have not yet been precisely identified, but some studies suggests that *HIV-1* gp 41 can be TLR10 ligand. A summary of ligands for Toll-like receptors is provided in Table 1.

### 2.2. Signal Transmission through TLRs

The studies performed so far have revealed how the ligand recognition signal is transmitted by TLRs. The fundamental step in the initiation of the signaling pathway is the formation of a dimer by the two extramembrane domains followed by the formation of a dimer by the TIR domains. TLR3, 4, 5, 7, 8, 9 receptors form homodimers, while heterodimers are formed by TLR1, 2, 6 (TLR1/2, TLR2/6) [46,58,59,60,61]. Each dimer binds different amounts of ligand. The TLR3 dimer binds a single dsDNA molecule, TLR1/2 and TLR2/6 heterodimers also bind one di- or triacetylated lipoprotein molecule each. In contrast, the TLR9 dimer attaches two fragments of CpG-rich DNA and each TLR4 dimer pair attaches two LPS molecules each [58,62].

The exact signaling pathways by which the information of microbial recognition triggers a cellular response and the production of substances to reduce and combat the existing threat are still being studied. Currently, most of the signal transduction pathway, the proteins involved in the signal transduction cascade, and the DNA fragments that respond to given signals have been established, but it is still possible to discover alternative pathways and new participants in this process. As we mentioned earlier, the signaling pathway begins with dimerization of intracellular TIR domains. TIR dimers attach one of the TIR-domain containing adaptor proteins, including MyD88, TRIF, TIRAP/MAL, or TRAM. MyD88 can be used by all TLRs to activate NF-kBs and MAPKs and subsequently stimulate the production of proinflammatory cytokines. TIRAP is a protein that is involved in the attachment of MyD88 to membrane TLRs such as TLR2 and TLR4 and it is directly involved in signal transduction by endosomal TLRs like TLR9. TRIF is attached to TLR3 or TLR4 and promotes an alternative signaling pathway leading to activation of IRF3, NF-kB, and MAPKs, causing increased synthesis of IFN I and proinflammatory cytokines. TRAM is connected to TLR4 as a bridge between TRIF and TLR4 and TLR3 connects directly to TRIF [63,64]. Depending on the adaptor attached, TLRs signaling pathways have been divided into two pathways, MyD88-dependent and TRIF-dependent.

#### 2.2.1. MyD88-Dependent Pathway

When MyD88 binds to TLRs, IRAK family proteins are attached and the resulting complex is called a myddosome. During myddosome formation, IRAK4 activates IRAK1, which binds to the RING-domain of E3 ubiquitin ligase TRAF6. TRAF6 with ubiquitin-conjugating enzyme UBC13 and UEV1A attached to it leads to K63-linked polyubiquitination of its own molecule and TAK1 kinase protein complex formation. TAK1 is a protein belonging to the MAPKKK family of proteins and forms a complex with three regulatory units, TAB1, TAB2, and TAB3, which react with polyubiquitinated chains generated by TRAF6, leading to TAK1 activation. TAK1 then leads to the activation of two pathways: the IKK-complex-NF-kB pathway and the MAPK-pathway. TAK1 phosphorylates IKKB of the IKK complex, resulting in the translocation of NF-kB to the cell nucleus and the stimulation of genes responsible for the production of proinflammatory cytokines. TAK1 activation also leads to the activation of MAPK family proteins such as Erk1/2, p38 and JNK, which lead to the activation of AP-1 family transcription proteins which stabilizes mRNA and leads to the regulation of inflammatory response [65,66,67].

#### 2.2.2. TRIF-Dependent Pathway

TRIF protein interacts with TRAF6 and TRAF3. TRAF6 recruits RIP-1 kinase, which in turn activates the TAK1 complex leading to activation of NF-kB and MAPKs and increased production of proinflammatory cytokines. In addition, TRAF3 recruits the kinases TBK1 and IKKi (related to IKK proteins), which interact with the NEMO protein during IRF3 phosphorylation. IRF3 then forms dimers that travel to the cell nucleus and cause increased gene expression for IFN I [65,66,67].

Pellino E3 ubiquitin ligases family proteins are also involved in the TRIF-dependent signaling pathway. Pellino-1 is phosphorylated by TBK1/IKKi which accelerates RIP-1 ubiquitination, suggesting that Pellino-1 mediates NF-kB activation in the TRIF-dependent pathway through RIP-1 recruitment [68].

#### 2.2.3. Balance between MyD88- and TRIF-Dependent Pathways

TLR4 activates both MyD88- and TRIF-dependent conduction pathways. Activation of these two pathways is under the control of several proteins that are responsible for an adequate response to detected infection. The balance in the production of proinflammatory cytokines and IFN I may have implications for the development of cancer and inflammatory and autoimmune diseases. TRAF3 is a protein found in both the myddosome and triffosome complex. In the MyD88 complex, TRAF3 is degraded, resulting in activation of TAK1. Additionally, TRAF3 functions as an inhibitor of the MyD88-dependent pathway. NRDP-1 E3 ubiquitin ligase binds and ubiquitinates MyD88 and TBK1, resulting in MyD88 degradation and TBK1 activation. This results in decreased production of proinflammatory cytokines and increased production of IFN I [69]. MHC class II molecules, which are localized in the endosomes of antigen-presenting cells, cause maintenance of Btk kinase activity. They use CD40 as a co-stimulatory molecule for this task. Activated Btk interacts with MyD88 and TRIF, causing the activation of both MyD88-dependent and TRIF-dependent signaling pathways, leading to enhanced production of proinflammatory cytokines and IFN I, respectively [70].

#### 2.2.4. Intrinsic and Pathogen-Related Negative Regulation of TLR Signaling Pathways

Negative regulation of signaling pathways is necessary to prevent excessive secretion of proinflammatory cytokines and IFN I, both of which may be involved in the development and exacerbation of inflammatory and autoimmune diseases. Targets for molecules that impair signal transduction occur at each key step in the signaling pathway. Activation of the MyD88-dependent pathway is inhibited by ST2825, SOCS-1, and Cbl-b, and the TRIF-dependent pathway is inhibited by SARM and TAG [71,72]. These proteins bind to MyD88 and TRIF preventing them from attaching to TLRs and initiating the signaling pathway. TRAF3 activation is inhibited by SOCS3 and DUBA and TRAF 6 is targeted by numerous inhibitors including A20, USP4, CYLD, and others [73,74]. Additionally, the transcription factor NF-kB can be inhibited by Bcl-3, IkBNS, Nurr1, ATF3, or PDLIM2 [75] and IRF3 activity is attenuated by Pin1 and RAUL [76].

Pathogens also have the ability to impair or inhibit TLR signaling pathways. These strategies can be divided into two categories. The first focuses on early steps in signaling pathways that are unique to a particular TLR and the second involves interference with steps in pathways that are not unique to a particular receptor, such as preventing activation of NF-kBs or MAPKs [75]. Examples include modifications in the conformation of bacterial LPS, whose new conformation is more difficult to recognize by the DC14-MD2-TLR4 complex [77,78]. Staphylococcus aureus changes the peptidoglycan structure to one that is more difficult to recognize in lysosomes [79,80]. Similarly, H. Pylori substitutes the amino acids encoding flagellin making recognition by TLR5 more difficult [52]. An increasing number of microorganisms encode TIR-domain-containing proteins in the genome that inefficiently bind to elements of the myddosome [75]. Some microorganisms produce enzymes that inactivate key signaling pathway proteins: e.g., viral TRIF degrading protease (*HCV, Coxsackie 3 virus*) [81].

## 3. NOD-Like Receptors (NLRs)

NOD-like receptors are another group of receptors after TLRs that belong to the PRRs family. They are present in cells of invertebrates and vertebrates, in humans 22 receptors of the NLRs family have been described so far [11,82]. These are receptors located in the cytoplasm of cells. The structure of NLRs is characterized by a common domain organization: (i) a centrally located nucleotide-binding NACHT domain (NAIP, HET-E, TP-20) that participates in autooligomerization and is essential for ATP-dependent activation of NLRs, (ii) an N-terminal effector domain that binds to adaptor proteins and downstream effectors to convey receptor excitatory information, and (iii) a C-terminal region that is composed of varying numbers of LRR repeats and is responsible for ligand recognition [83,84]. Human NLRs have been divided into four subgroups, depending on the structure of their N-terminal domain, which may include (i) acidic transactivation domain (AD)-subgroup NLRA, (ii) baculoviral inhibitory repeat-like domain (BIR)-subgroup NLRB, (iii) caspase activation and recruitment domain (CARD)-subgroup NLRC, or (iv) pyrin domain (PYD)-subgroup NLRP [84,85].

The NLRA subgroup includes only one type of receptor called CIITA (Class II Histocompatibility Complex Transactivator). The C domain of this receptor contains four LRR repeats and a GTP-binding domain. GTP attachment accelerates transport of the molecule to the cell nucleus, where it plays a role in activating gene expression for MHC class II not by binding to DNA but by intrinsic acetyltransferase activity [86,87,88]. The NLRB-subgroup also includes only one receptor, NAIP (NLR Family Apoptosis Inhibitory Protein) For a long time, NAIP was considered as an anti-apoptotic protein that acts by inhibiting the activity of caspase 3 (CASP3), CASP7 and CASP9, but now, NAIP is consider rather as a sensor of bacterial flagellin or type-3 secretion system components delivered by pathogens leading to NLRC4 inflammasome activation. NAIP mediates also neuronal survival in various pathological states and protects against apoptosis induced by a variety of factors [89,90,91]. NLRC is a subgroup involving 6 receptors, nucleotide oligomerization domain 1 (NLRC1) (NOD1), nucleotide oligomerization domain 2 (NLRC2) (NOD2), NLRC3, NLRC4, NLRC5, and NLRX1. NOD1 and NOD2 are considered the two major receptors belonging to the NLRC. They recognize intra-bacterial building blocks that enter the cell through direct bacterial invasion or through other cellular uptake mechanisms [92,93]. The NLRP family includes 14 receptors characterized by the presence of a PYD domain at the N-terminus that is responsible for transmitting an pyroptotic signal or inducing an inflammatory response [94,95]. Interestingly, studies have shown that the NLRP family of receptors plays a significant role in both innate immunity and reproduction in mammals. It has been suggested that NLRPs may play a role in oogenesis and early stages of embryogenesis (pre-implantation) [96,97].

As PRRs, NLRs recognize a wide range of PAMPs, such as bacterial cell wall components (peptidoglycan, flagellin), microbial secreted toxins, viral RNA, fungal sharps, or even entire microbial and parasitic organisms [10,98,99,100]. DAMPs that are recognized by NLRs include: ATP, hyaluronic acid, sodium urate (MSU), uric acid, and cholesterol crystals [6,7,101]. Moreover, environmental substances such as asbestos, silicon, aluminum, skin irritants, and UV radiation can cause stimulation of NOD-like receptors. A summary of ligands for NLRs is provided in Table 2. However, not all NLRs function as PRRs as some respond to the presence of cytokines such as interferons.

Activated NLRs play various roles that can be divided into four groups: inflammasome formation, signal transduction, stimulation of gene transcription, and autophagy [93,102,103,104,105,106].

### 3.1. Formation of Inflammasomes

Inflammasomes are multi-molecular complexes that activate caspase-1. Activated caspase-1 leads to the maturation of pro-interleukin-1B (pro-IL-1B) and pro-interleukin-18 (pro-IL-18) into their active forms IL-1B and IL-18 [103,104]. A consequence of inflammasome formation may also be the phenomenon of pyroptosis, a term used to describe inflammatory, programmed cell death dependent on caspase 1/4/5/11. In addition, pyroptosis leads to the release of DAMPs and an enhanced inflammatory response [107,108]. Receptors capable of forming inflammasomes include NLRP1, NLRP2, NLRP3, NLRP6, NLRP7, NLRP12, and NLRC4 (after activation by NAIP) [103,109]. Inflammasome formation is a response to recognition by PAMPs receptors of such things as bacterial toxins, flagellin proteins, muramyl dipeptide, viral or bacterial RNA and DNA, fungal fragments, fungal mannan, zymosan, and protozoa-derived hemozoin. Sterile activators include DAMPs such as ATP, cholesterol crystals, hyaluronic acid, sodium urate, or amyloid and environmental factors include silicon, aluminum, asbestos, UV radiation, and others [103]. The structure of the inflammasome is composed of a receptor (NLR), an ASC (apoptosis-associated speck-like protein containing CARD) adaptor molecule, and an effector molecule (pro-CASP1). Upon ligand recognition, NLR attaches the ASC molecule via pyrin-pyrin domain binding. Subsequently, pro-CASP1 binds to ASC via the CARD-CARD domain resulting in the formation of a functional inflammasome [110]. NLRP1 has a CARD domain that can directly bind to pro-CASP1 without the involvement of ASC [111,112] protein whereas NLRC4, which lacks the PYD domain, can form two types of inflammasomes. The attachment of ASC protein to NLRC4 results in increased secretion of IL-1B and IL-18 and lack of ASC attachment during inflammasome formation results in pyroptosis [113,114]. The ligand for NLRC4 located in macrophages may be Salmonella rods [115]. NAIP and NLRC4 form the NAIP-NLRC4 inflammasome upon recognition of the bacterial flagellin and bacterial type III secretion system [114]. NLRP1 inflammasomes are formed after recognition of muramyl dipeptide (MDP), a common bacterial peptidoglycan (produced by G+ and G− bacteria) and after exposure to anthrax toxins [111,112]. NLRP3 forms inflammasomes under the influence of G+ bacteria, viruses, fungi, protozoa, ATP, and ROS. In addition, mitochondrial DNA and RNA can also activate NLRP3 [95,116]. NLRP7 can recognize bacterial lipopeptides [117].

### 3.2. Signal Transduction

The NOD1 receptor recognizes peptidoglycan derived from G- bacterial D-y-glutamyl-meso-diaminopimelic acid (iE-DAP) and the NOD2 receptor recognizes bacterial G- and G+ MDP [118]. Upon binding to ligand, those NLRs bind to receptor-interacting serine/threonine-protein kinase 2 (RIPK2), resulting in the formation of the RIPK2-IkB complex. Subsequently, RIPK2 causes activation of TAK1 kinase, which is a prerequisite for IKK complex activation and MAPK pathway activation. IKK-dependent dephospho-rylation of the NF-kB inhibitor (IkappaBalpha) enables its translocation to the cell nucleus and the associated increased transcription of genes for pro-survival cytokines. Activation of the mitogen-activated protein kinase (MAPK) pathway also results in increased secretion of proinflammatory cytokines by the cell [118].

In contrast to NOD1 and NOD2, some NLRs play a role as inhibitors of intracellular signaling pathways mediated by different receptors. Such effects have been described, among others for NRLC3 in T cells, NLRC5 in various non-specific immune cells, NLRP2/NLRP4 which are negative regulators of the NF-kB pathway through TRAF6-modifying effects, NRLP7 inhibits IL-1B secretion and NLRP10 inhibits caspase-1 activation [119,120,121,122,123,124,125].

### 3.3. Activation of Transcription

Another example of the diversity of functions of NLRs in the body is the CIITA receptor (NLRA). Several experiments found that NLRA fully corrected the defect in MHC class II production in cells from patients suffering from bare lymphocyte syndrome (BLS), a severe immunodeficiency. CIITA acts as a transactivator of gene expression for MHC class II. CIITA acts as a scaffold that allows the recruitment of DNA-binding transcription factors and histone-modifying enzymes to the promoters of MHC class II genes [126]. The mechanism of this action is not yet well understood, nor is CIITA’s function as a PRR.

Another receptor of the NLRs-NLRC5 family has an important role in the transcription of genes for MHC class I. IFN-y-treated NLRC5 acts as a transactivator of gene expression by folding regulatory factor X (RFX), c-AMP-responsive-element binding protein 1 (CREB1), activating transcription factor 1 (ATF1), and nuclear transcription factor Y (NFY) into the SXY module in the promoter of genes for MHC class I. (Footnote to NLRC5 in MHC I). Although the expression of genes for MHC class I and II depends on several transcription factors (such as NF-kB, IFN regulatory factor family, RFX, CREB1, ATF1 or NFY), the presence of CIITA and NLRC5 is essential for this process to take place [83].

### 3.4. Autophagy

Autophagy is one of the primary mechanisms for maintaining homeostasis, in which a cell digests elements of its structure in order to permanently remove them or return certain components to metabolic turnover. Autophagy is also a way to defend the body against intracellular microorganisms including bacteria, viruses, and protozoa, and has a regulatory function in the immune response. The process of autophagy occurs through the formation of unique organelles called autophagosomes. Autophagosomes fuse with lysosomes resulting in the destruction of structures contained in autophagosomes. Many mediator proteins are involved in autophagosome formation and fusion with lysosomes, mammalian target of rapamycin complex 1 (mTORC1), AMP-activated protein kinase (AMPK), serine/threonine protein kinase (ULK1), class III phosphatidylinositol-3-phosphate kinase (PI3K) complex, mammalian homolog of autophagy related proteins (ATG8), and others [105,106,127].

Xenophagy is a type of autophagy in which elements of the autophagosome are intracellular pathogens and other substances foreign to the body. The mechanism of xenophagy is similar to classical autophagy but requires ligand recognition by the PRR. NOD1 and NOD2 recognize bacterial ligands and initiate xenophagy by recruiting ATG16L1. NLRX1, which is localized in mitochondria, regulates autophagy associated with viral infection by interacting with Tu translation elongation factor of mitochondria (TUFM), which later reacts with ATG5-ATG12 and ATG16L1 [105,106,127].

## 4. RIG-I-Like Receptors (RLRs)

RIG-I-like receptors are cytoplasmic nucleic acid receptors for RNA virions, RNA replication intermediate molecules, and transcription products (footnote to RLRs 1). The RLRs family includes three receptors—retinoid acid inducible gene I (RIG-I), melanoma differentiation factor 5 (MDA5), and laboratory of genetics and physiology 2 (LGP2). They are located in the cytoplasm of most cells in the body, although recent studies report that RLRs may also be located in the cell nucleus [128]. RLRs belong to the DexD/H box helicase family, which is in turn part of the type 2 helicase superfamily. RLRs are made of (i) a conserved helicase core that contains two helicase domains Hel1 and Hel2 separated by a fragment known as Hel2i, (ii) a C-terminal domain (CTD), (iii) RIG-I and MDA5 have an additional two CARD domains. The core domain and CTD interact to detect RNA immunostimulators and CARD is responsible for transducing the ligand recognition signal [9,129]. LGP2, which lacks a CARD domain, is thought to be a receptor that regulates RIG-I and MDA5. It inhibits the RIG-I receptor and enhances the response elicited by MDA5 [130,131]. It is now known that viral infections caused by the vast majority of viruses are recognized by RLRs. A summary of ligands for RLRs is provided in Table 3.

### 4.1. Ligands for RLRs

The structures that are recognized by the RIG-I receptor are 5′-triphosphate-RNA (5′-ppp-RNA) and 5′-diphosphate-RNA (5′-pp-RNA) found in both ssRNA and dsRNA, which are not present in most mature cellular RNA. In addition, the 5′-triphosphate and 5′-diphosphate fragments must have adjacent neighboring structures paired in conformations such as a hairpin. They are found in the RNA structure of most viruses, e.g., influenza A virus (IAV). Specific sequences such as poly-U or poly-UC present in the genome of, for example, hepatitis C virus (HCV) are also recognized by RIG-I. Studies have shown that RIG-I recognizes genomic RNA as effectively as RNA of damaged interfering Sendai virus and IAV molecules. The ligand for RIG-I may also be the 5′E-region of hepatitis B virus (HBV) pre-genomic RNA [132,133,134,135,136,137,138]. Ligands for the MDA5 receptor are much less well understood than those for RIG-I. MDA5 is known to detect large RNA aggregates produced in cells during encephalomyocarditis virus (EMCV) infection. These aggregates contain both ssRNA and dsRNA. Furthermore, ribose 2′-O methylation in the 5′-cup structure avoids MDA5 activation [132,139,140]. Both receptors have the ability to recognize a synthetic dsRNA ligand—polyinosilic-polycytidylic acid (poly(I:C)) but RIG-I recognizes shorter dsRNA fragments and MDA5 responds to high molecular weight poly(I:C)- HMW-poly(I:C) [9,141]. It has recently been elucidated how host cellular RNA can be a ligand for RLRs receptors during viral infection. In studies of cells infected with herpes simplex virus 1 or Kaposi’s sarcoma-associated herpesvirus (KSHV), it was found that the RIG-I receptor is associated with pseudogene 5S rRNA-mainly *RNA5SP141* transcription products. The biological role of these 5′-ppp-RNA-containing transcripts is unknown. HSV-1 infection uncovers cellular non-coding RNAs in two ways. First, transcription products of the 5S rRNA pseudogene are inappropriately relocalized to the cytoplasm of the cell. Secondly, HSV-1 infection causes “shutdown” of the mRNA encoding the *RNA5SP141* binding proteins TST and MRPL18. Similar mechanisms may occur with infections with other viruses of the herpesvirus group, such as *Epstein-Barr virus* [142,143].

### 4.2. Signal Transduction through RLRs

In healthy, uninfected cells, RIG-I and MDA5 receptors are constitutively phosphorylated at several specific serine and threonine residues in the CTD and CARD domains, which keeps them in a state of suppressed signal transduction. Furthermore, RIG-I is maintained in an auto-repressed conformation due to interactions between the helicase domain and the CARD domain, whereas MDA5 presents an open conformation without the presence of foreign RNA [144]. Upon RNA binding, RLRs undergo a conformational change dependent on their ATPase activity a which is stimulated by binding to PACT (protein kinase R activator) [145]. These conformational changes trigger the release of CARD domains, which then bind to regulatory molecules. RIG-I and MDA5 bind to the phosphatase PP1-isoform PP1A or PP1y, which causes dephosphorylation of CARD domains. The RIG-I receptor then attaches TRIM (tripartite motif protein 25) and Riplet (also known as RNF135) proteins, which attach Lys63- linked ubiquitin polymers to the CARDs and C-terminal domain, respectively. The attachment of these polymers is essential for the tetramerization process of the RIG-I receptor and its interaction with the MAVS adaptor protein located on the outer membrane of mitochondria and mitochondria-associates membranes (MAMs) [146]. MDA5 binds dsRNA using Hel-CTD domains and forms filaments along the entire length of dsRNA. Unlike RIG-I, MDA5 does not bind dsRNA ends and filament formation is necessary for stable dsRNA binding and activation of the signal transduction pathway. The filaments formed by MDA5 then bind to MAVS [147]. Once stimulated, MAVS begins to form a prion-like filament structure that is the initiator to the formation of a large signaling complex composed of TRAF (Tumor necrosis factor Receptor-Associated Factors) proteins andTBK1 (or IKKE (IkB kinase-E)) The IKK-a–IKK-B–IKK-y triple complex is activated on the ubiquitin chains anchored to the TRAFs. This in turn leads to activation of IFN regulatory factor 3 (IRF3) and/or IRF7 and NF-kB. IRF3, IRF7, and NF-kB, with the participation of AP1 (activator protein 1), lead to increased transcriptional activity of genes for IFNs and other cytokines: TNF, IL-6, and IL-8. IFNa/B secretion then leads to increased transcription of a significant number of ISGs (interferon stimulated genes) resulting in the creation of an “antiviral environment” in virus-infected cells and in adjacent healthy cells [148]. Interestingly, RIG-I and MDA5 have the ability to block viral replication directly by inhibiting the binding of viral proteins to viral RNA [149].

### 4.3. RLR Regulation

Proper regulation of RLRs is essential for homeostasis—it ensures fast and effective response to viral infection and protects against excessive inflammatory response. Both positive and negative regulatory proteins are involved in the regulation of RIG-I and MDA5 receptors. ZAPS protein binds to RIG-I, causes its oligomerization and enhances its ATPase properties leading to increased IFN I production [150]. TRIM25 accelerates K63-linked polyubiquitination in the CARD domain and enhances the interaction between RIG-I and MAVS. The Riplet protein has a similar effect, but it causes ubiquitination in the CTD domain [146]. Ubiquitin specific protease 15 (USP15) prevents LUBAC-dependent degradation of TRIM25 which also promotes RIG-I signaling pathways [151]. Ubiquitination, which is essential for the early steps of signal transduction by RLRs, is a reversible process. The deubiquitinases USP3 and USP21 inhibit RIG-I activity by removing K63-linked ubiquitin chains. RIG-I is also inhibited by RNF125 and Siglec-G-/SHP2/c-Cbl- mediated K48-linked ubiquitination and protein degradation [152]. Furthermore, RIG-I is inhibited by Ser-Thr phosphorylation in the CARD domain [153]. Much less is known about the regulation of MDA5 activity. DAK, a dihydroacetone kinase, binds to MDA5 (but not to RIG-I) and specifically inhibits MDA5-linked IFN I production [154]. USP3 deubiquitinase reduces MDA5 activity by deubiquitinating its CARD domain [152].

MAVS is a key factor in the signal transduction pathway through RLR receptors. NLRX1 blocks MAVS and RIG-I binding and thus reduces IFN I production. The autophagy proteins Atg5 and Atg12 have an inhibitory effect on the RIG-I signaling pathway through interactions with RIG-I and MAVS. The stability of the MAVS complex is regulated by several E3 ligases. PCBP2 was identified as a MAVS-degrading protein in association with the HECT domain-containing E3 ligase AIP4. Smurf2 and TRIM25 are also negative regulators of the RLRs pathway as they degrade MAVS through K48-linked ubiquitinization [155,156,157].

## 5. Clinical Significance of TLRs, NLRs, and RLRs

The effect of over-stimulation of TLRs, NLRs, and RLRs on the development of autoimmune diseases and chronic inflammation has been known for many years. A description of these relationships is beyond the scope of this paper, readers are referred to other studies [4,158,159]. In addition, prolonged inflammation is an ideal environment for the development of cancer. On the other hand, immune surveillance is necessary to detect abnormal, cancerous cells and prevent their proliferation [160]. Nowadays, it is known that in the tumor microenvironment (TME) not only stromal cells, fibroblasts, and endothelial cells are present, but also cells of the immune system. So far, the attention of researchers has been focused mainly on T lymphocytes belonging to the specific immune system. Nowadays, much attention is paid to non-specific immune cells such as macrophages (called tumor-associated macrophages, TAM), dendritic cells, neutrophils, NK cells, myeloid derived suppressor cells (MDSCs), and innate lymphoid cells (ILCs). Proinflammatory substances secreted by these cells contribute to genetic instability, promote angiogenesis, and facilitate metastasis of cancer cells. In addition, they suppress the body’s immune response and, once treatment is initiated, cause the tumor to be more resistant to chemotherapy and immunotherapy [14,159,161]. At the same time, increasing sensitivity to tumor cell-derived antigens and activating the nonspecific immune system by ligand-PRR interaction may be a strategy to avoid tumor escape from immune surveillance [162]. Immunotherapy strategies aimed at activating cells of the innate as well as acquired immune system are also effective in eradicating large tumor masses [163,164], not uncommon at diagnosis or recurrence of hematopoietic proliferative disease.

### 5.1. Ligands for TLRs in the Treatment of Hematopoietic and Lymphatic Diseases

Activation of Toll-like receptors leads to enhanced anti-tumor responses by several mechanisms including increased secretion of proinflammatory cytokines in the tumor microenvironment to induce immune responses by innate and acquired mechanisms and induction of apoptosis or necrosis of tumor cells [165]. Moreover, activation of TLR signaling pathways leads to maturation of antigen-presenting cells—macrophages and dendritic cells, increased production of IFNs I by these cells, increased expression of CD80, CD86, and CD40 molecules, which subsequently activate other cells of the innate immune system, as well as tumor-specific T-cell responses [166,167,168]. The cytokines IL-6 and IL-12 are the most important in cancer immunotherapy with TLR receptor ligands. IL-6 enhances antigen-specific T cell activation by inhibiting Treg cells, IL-12 promotes a Th1-directed response profile [169,170,171].

BCG (Bacillus Calmette-Guerin) is a substance that activates TLR2 and TLR4 through mycobacterium components and TLR9 through the presence of bacterial DNA. It has been used in combination with standard chemotherapy to treat patients with acute myeloid leukemia. Single reports also demonstrate the efficacy of BCG in the monotherapy of AML [172,173]. Poly I:C, poly ICLC molecules that bind to the TLR3 receptor are being investigated as adjuvant molecules in the development of cancer vaccines. In hematological malignancies, the indication is therapy for NHL [9,141,174]. In addition, poly ICLC in combination with radiotherapy is being investigated as a treatment option for cutaneous T-cell lymphoma and in combination with the rhuFlt3L/CDX-301 molecule for the treatment of low-grade B-cell lymphoma [174]. LPS, a TLR4 receptor ligand, is being studied as an ex vivo stimulant for dendritic cells in the treatment of NHL. Another TLR4 ligand-G100-in combination with pembrolizumab is being studied for efficacy in the treatment of follicular lymphoma. TLR7- 852A receptor ligand has been investigated as an immunotherapy for hematological malignancies including AML, ALL, NHL, HL, and MM [175]. The TLR8 receptor binds to a synthetic molecule, VTX-2337, and in combination with radiation therapy may have applications in the treatment of low-grade B-cell lymphomas. The most widely used in immunotherapy of hematopoietic and lymphoid malignancies are CpG ODNs, single-stranded oligodeoxynucleotides, characterized by the presence of repeats containing cytosine and guanine. CpG 7909 is a TLR9 ligand whose efficacy has been studied in the treatment of cutaneous T-cell lymphoma and NHL and in second-line therapy for patients with chronic lymphocytic leukemia [176]. The use of other CpG molecules is being widely studied as a means of immunotherapy for chronic lymphocytic leukemia. CpG-treated leukemic B lymphocytes undergo apoptosis [177,178]. The efficacy of TLR9 ligands in combination with radiotherapy or as part of combination treatment with targeted therapy, monoclonal antibodies or cytokines is under investigation for the treatment of NHL [179]. Most of the studies mentioned are in phase I/II clinical trial [165].

### 5.2. Ligands for NLRs in the Treatment of Hematopoietic and Lymphoid Malignancies

NLR receptor signaling pathways may be involved in the development of cancer, but also may be used to treat it. In humans, it has been observed that in some solid tumors (e.g., lung cancers, breast cancers, head and neck epithelial cancers), there is a greater expression of NOD1 and NOD2 on tumor cells and greater polymorphism for NOD2 [14,158]. Increased inflammasome activity can also lead to cancer development. Mutations in the gene for NLRP3 that cause persistent stimulation of this receptor have been implicated in melanoma susceptibility, colorectal cancer prognosis, and overall survival in multiple myeloma [180]. The phenomenon of pyroptosis may show a strong correlation with tumor cell proliferation and migration in various types of cancer. Pyroptosis causes inflammatory death of tumor cells, thereby inhibiting tumor growth and its ability to metastasize. Molecules that promote inflammasome formation and pyroptosis phenomenon are currently being investigated for use as part of anticancer treatment. These substances include non-coding RNA and other elements recognized by NLRs receptors [108,181]. An example of an investigational substance recognized by NLRP3 is anthocyanin, showing efficacy in the treatment of hepatocellular carcinoma and oral squamous cell carcinoma [182,183]. However, little is known about their use in the treatment of hematologic malignancies.

### 5.3. Ligands for RLRs in the Treatment of Hematopoietic and Lymphoid Malignancies

The death of a virus-infected cell is an important mechanism that leads to the elimination of diseased cells and prevents the spread of infection. Activation of RLRs receptors by viral or synthetic ligands leads to production of IFN I and also activation of ISGs and direct cell death or to induction of apoptosis. Furthermore, cancer cells are highly susceptible to RLR-dependent apoptosis while non-cancer cells are resistant to it through endogenous Bcl-xL [184]. Bhoopathi et colleagues in their studies revealed that the immune response against viruses and cancer cells follows similar pathways; therefore, they decided to use RLRs ligands as anticancer substances [184]. Several 5′ppp-siRNA molecules have been developed to activate RLR signaling pathways, and also to simultaneously block oncogenes or immunosuppressive pathways in cancer cells. Application of 5′ppp-siRNA for Bcl-2 in malignant melanoma resulted in inhibition of tumor growth due to, among other things, downregulation of Bcl-2. In contrast, the use of 5′ppp-siRNA for TGF B has been studied in the treatment of pancreatic cancer. Activation of the MDA5 receptor by the synthetic dsRNA-poly I:C molecule in ovarian cancer resulted in increased cell surface MHC I expression, enhanced secretion of IFN I and other cytokines, and enhanced cell apoptosis [13]. Another important aspect of the use of RLR ligands in cancer therapy is that apoptosis induced by RLRs receptors is independent of the mutational status of the p53 gene, mutations of which are responsible for the resistance of cancer cells to chemotherapy and radiotherapy-induced apoptosis. Research into the use of ligands for RLRs is ongoing among patients with advanced solid tumors refractory to standard treatment. A summary of ligands for PRRs potentially useful in the treatment of hematopoietic diseases is provided in Table 4.

## 6. Conclusions

Innate immune mechanisms are the body’s first line of defense against infection. Over many years of research, the structure, functions, and modes of action of many families of PRRs have been studied in detail. However, their role is still not fully understood, especially in the field of using receptors and their ligands in cancer treatment. So far, it is known that TLRs, NLRs, and RLRs show great potential in immunotherapy of solid organ cancers as well as hematological malignancies. However, further research into their introduction in treatment regimens is warranted.

## Figures and Tables

**Table 1 ijms-22-13397-t001:** Receptor ligands for TLRs; dsRNA-double-stranded RNA, ssRNA-single stranded RNA, LPS-lipopolisacharide, and dsDNA-double-stranded DNA.

TLR	Location in the Cell	Ligand	Origin of the Ligand
TLR1/2	Cell membrane	Triacylated lipopeptides	Bacteria, Mycobacteria
TLR2	Cell membrane	Hemagglutinins, glycosylphosphatidylinositol, phospholipoman, lipoarabinomannan, peptidoglycans, porins, lipoproteins	Bacteria, Mycobacteria, viruses, fungi, parasites, self
TLR3	Endolysosomal membrane	dsRNA, ssRNA	Viruses
TLR4	Cell membrane	LPS, mannan, inositol phospholipids, envelope proteins	G- bacteria, viruses, self
TLR5	Cell membrane	Flagellin	Bacteria
TLR2/6	Cell membrane	Diacylated lipopeptides, zymosan, lipoteichoic acid	Bacteria, mycobacteria, viruses, fungi
TLR7 (and human TLR8)	Endolysosomal membrane	ssRNA	Viruses, bacteria, fungi
TLR9	Endolysosomal membrane	dsDNA, CpG DNA, hemozoin	Bacteria, viruses, protozoa, self
TLR10	Endolysosomal membrane	HIV-1 gp41	Viruses
TLR11 (mice)	Cell membrane	Profilin-like molecules	Protozoa

**Table 2 ijms-22-13397-t002:** Receptor ligands for NLRs and their functions in the body [84,102]. PRR-pattern recognition receptors, DAP-diaminopimelic acid, MDP-muramyl dipeptide, ROS-reactive oxygen species, and DAMPs-damage associated molecular patterns.

Subgroup	NLR	Ligand/Function
NLRA	CIITA	Regulation of MHC II expression
NLRB	NAIP	PRR for flagellin, pyroptosis, inhibition of apoptosis
NLRC	NOD1	PRR for DAP
	NOD2	PRR for MDP, viral ssRNA, autophagy,
	NLRC3	Negative regulation of T cell activation and TLR activation
	NLRC4	PRR for flagellin, rod proteins, pyroptosis, phagosome formation
	NLRC5	Upregulation of MHC I expression, regulation of innate response
	NLRX1	ROS production, autophagy induced by viral infection
NLRP	NLRP1	PRR for MDP and anthrax toxin
	NLRP2	Negative regulation of NF-kB, embryonic development
	NLRP3	PRR for DAMPs
	NLRP4	Negative regulation of IFN I, autophagy
	NLRP5	embryogenesis
	NLRP6	Negative regulation of NF-kB
	NLRP7	PRR for lipopeptides
	NLRP8	unknown
	NLRP9	unknown
	NLRP10	Migration of dendritic cells
	NLRP11	unknown
	NLRP12	Negative regulation of NF-kB
	NLRP13	unknown
	NLRP14	spermatogenesis

**Table 3 ijms-22-13397-t003:** Receptor ligands for RLRs.

Virus Families (Examples)	RLR
Herpesviridae (Herpes simplex virus 1, Epstein-Barr virus, Kaposi’s sarcoma-associated herpesvirus)	RIG-I, MDA5
Poxviridae (vaccinia virus)	RIG-I, MDA5
Adenoviridae (adenoviruses)	RIG-I
Reoviridae (rotavirus)	RIG-I, MDA5
Picornaviridae (rhinovirus, coxsackie B)	RIG-I, MDA5
Flaviviridae (HBV, Zika virus)	RIG-I, MDA5
Coronaviridae (SARS coronavirus)	RIG-I, MDA5
Orthomyxoviridae (Influenza A virus)	RIG-I
Paramyxoviridae (measles virus)	RIG-I, MDA5
Filoviridae (Ebola virus)	RIG-I, MDA5
Retroviridae (HIV)	RIG-I, MDA5
Hepadnavridae (HBV)	RIG-I, MDA5

**Table 4 ijms-22-13397-t004:** Ligands for PRRs in the treatment of hematopoietic diseases. References in the text.

Group of PRRs	PRR	Ligand	Hematopoietic Disease
**TLR**	TLR2 TLR4 TLR9	BCG	Acute myeloid leukemia
	TLR3	polyI:C, polyICLC	Non-Hodgkin lymphomas, especially cutaneous T-cell lymphoma, low-grade B-cell lymphoma
	TLR4	LPS	Non-Hodgkin lymphomas
	TLR4	G-100	Follicular lymphoma
	TLR7	852A	Acute myeloid leukemia, acute lymphoblastic leukemia, Non-Hodgkin lymphomas, Hodgkin lymphoma, Multiple myeloma
	TLR8	VTX-2337	Low-grade B-cell lymphoma
	TLR9	CpG 7909	Non-Hodgkin lymphoma, especially cutaneous T-cell lymphoma, Chronic lymphocytic leukemia
**NLR**	NLRP3	anthocyanin	Non-hematological malignancies: hepatocellular carcinoma, oral squamous cell carcinoma
**RLR**	RIG-I	5’ppp-siRNA for Bcl-2	Non-hematological malignancies: malignant melanoma
	RIG-I	5’ppp-siRNA for TNF-β	Non-hematological malignancies: pancreatic cancer
	MDA5	dsDNA-poly I:C	Non-hematological malignancies: ovarian cancer

BCG- Bacillus Calmette-Guerin, LPS- lipopolysaccharide.

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
