# Peer review of "Toll-Like Receptors (TLRs), NOD-Like Receptors (NLRs), and RIG-I-Like Receptors (RLRs) in Innate Immunity. TLRs, NLRs, and RLRs Ligands as Immunotherapeutic Agents for Hematopoietic Diseases"

_ijms, 2021, doi:10.3390/ijms222413397_

Round 1
Reviewer 1 Report
In this manuscript, the authors present an overview of the TLRs, RLRs and NLRs as well as the use of their respective ligands as therapeutic agents against hematopoietic diseases.
This review is quite complete and suitable for publications. I have however few comments.
Line 48. Genetic material of bacteria is also a PAMP
Line 57. STING is not a receptor of PAMPs per se, it is only an adaptor like MAVS in the RLR pathway.
Line 53. 6 groups of PRRs have been indeed described but here only 5 are mentioned (TLRs, RLRs, NLRs, CLRs and cGAS-STING). AIM2-like-receptors (ALRs) should be then also cited.
Line 188. TAK1 actually phosphorylates IKKB of the IKK complex for activation.
Line 266. NAIP is now more accepted as a sensor of flagellin leading to NLRC4 inflammasome activation rather than an inhibitor of apoptosis.
Line 277. The PYD domain of NLRPs is more required for transmitting a pyroptotic signal rather than an apoptotic signal, don’t you think?
Line 303. NLRC4 forms an inflammasome after activation by NAIP. NAIP does not form inflammasome.
Line 329. “those” is missing before NLRs.
Line 332 and 433. Dephosphorylation not defosphorylation
Line 333. NF-KappaB inhibitor (IKappaBalpha)
Line 444. There is a mistake, TNF is not in the MAVS complex. I think that the authors meant TRAFs (Tumor necrosis factor Receptor-Associated Factors) which are found in complex with MAVS.
Line 445. I’m not sure that the IKK complex (IKKalpha, beta and Gamma) is really in complex with MAVS but is activated on the ubiquitin chains anchored to the TRAFs. IKK Epsilon and IKKi (written somewhere else in the text) are the same protein, a homolog of TBK1.
Line 565. I do not understand the sentence “therefore in this study we decided to use RLRs ligands”. Which study are the authors talking about?
Line 567. Immunosuppressive.
Reviewer 2 Report
The manuscript by Wicherska-Pawlowska et al. is a comprehensive review on the role of pattern-recognition receptors (PRRs) in the mammalian immunity. This is a current and important topic in the contemporary immunology area. The manuscript is well-written and appropriately organized. It firstly provides an information regarding the roles of the representatives of main families of PRRs in the functioning of the immune system and inflammatory processes, and then describes the most promising clinical applications of the agents modulating the functions of PRRs, with special emphasis put on the cancer treatment.
There are some minor comment regarding the manuscript:
- Lines 91-92: “In mammals, 12 genes encoding TLRs have been identified so far (10 in humans and 12 in mice)” – this statement needs to clarified as there are reports stating that human genome contains indeed 10 TLR-encoding genes (if the pseudogene for TLR11 is not counted), and in the mouse there are 12 TLR-encoding genes (TLR1 to TLR13, but without TLR10). Altogether it would sum up to 13 different TLR-encoding mammalian genes (TLR1-13).
- Line 140: “b-defensins” – should be “beta-defensins”
- Lines 147-148: “Ligands for the human TLR10 receptor have not yet been identified” - more information would be expected on this subject, as there are some recent reports regarding the putative ligands and functions of TLR10 (e.g. https://www.frontiersin.org/articles/10.3389/fimmu.2019.00482/full or https://onlinelibrary.wiley.com/doi/full/10.1111/sji.12988 )
- Section 5: a table summarizing the most prominent examples of PRR-modulating agents in the clinical use could be helpful in this section.
